# What You See Is What You Get: Entity-Aware Summarization for Reliable Sponsored Search

**Xiao Liang**[1,3][*], **Xinyu Hu**[2][*], **Simiao Zuo**[2], **Jimi He**[2], **Yu Wang**[2], **Victor Ye Dong**[2],
**Yeyun Gong**[3][†], **Kushal S. Dave**[2], **Qiang Lou**[2], **Yi Liu**[2], **Shao-Lun Huang**[1], **Jian Jiao**[2][†]
[1]Tsinghua University  [2]Microsoft AI  [3]Microsoft Research
liangx22@mails.tsinghua.edu.cn

## Abstract

Large Language Models (LLMs) are increasingly used to generate summaries for sponsored search results, but they often misalign with actual webpage content, leading to user misinformation and retrieval inaccuracies. Existing approaches often fail to accurately capture critical entity information and effectively determine query-document relevance, limiting their effectiveness in sponsored search contexts. We propose an entity-aware summarization framework to improve the reliability of AI-generated summaries for sponsored search. Our approach involves two key steps: (1) a structured process for generating entity-aware summaries, including webpage entity tagging, query reflection, and summary generation; and (2) fine-tuning LLaMa3.1-8B on entity-rich summaries and applying Direct Preference Optimization (DPO) to enhance query-document relevance. Comprehensive evaluations demonstrate the superiority of our method over existing baselines. We achieve F1 scores of 0.57, 0.44, and 0.26 for Brand, Product, and Feature entities, respectively, and show a 7.86% improvement in recall@50 for retrieval tasks. Our approach significantly improves the alignment between AI-generated summaries and webpage content in sponsored search environments, marking an important advancement in accurate and effective AI-driven information retrieval systems.

## 1   Introduction

Sponsored search is a fundamental driver of revenue for commercial search engines, connecting users with products and services through targeted advertisements on search engine result pages (SERPs). A key element of these sponsored results is the brief summary of webpage content displayed to users, which plays a crucial role in influencing click-through rates and user engagement. Large Language Models (LLMs) have the potential to transform this process by generating compelling, AI-driven summaries for sponsored search results directly within SERPs [9]. However, a significant challenge remains: the risk of misalignment between these AI-generated summaries and the actual content of the referenced webpages [19]. For instance, an AI might summarize a webpage comparing prices of different Apple Watch series as *"Check out all different Apple Watches"* misleadingly suggesting purchase information. When users click through expecting to see buying options but find only price comparisons, their expectations are unmet, leading to dissatisfaction. A more entity-aware summary would highlight the *price comparison* service.

Such misalignment can lead to misinformation, erode user trust, and negatively impact user experience, which is crucial for both advertisers and search platforms. Two primary issues complicate the generation of accurate SERP summaries. **Lack of entity awareness:** despite their advanced capabilities, LLMs often struggle to accurately represent critical entities such as brands, products, or features, which are essential for aligning summaries with webpage content. **Ineffective query-document**

---

[*]Equal contribution. This work is done during Xiao Liang's internship at Microsoft Research.
[†]Corresponding authors.

**relevance:** Product and service webpages typically cover multiple items or even brands, making it challenging to establish relevance between specific user queries and the content. Current methods lack mechanisms to generate dense, precise representations that can capture this relevance effectively.

To address these challenges in SERP summary generation, we propose an entity-aware summarization framework designed to enhance the reliability of AI-generated summaries for sponsored search results. Our framework introduces a structured, three-step process specifically tailored for SERP content:

First, we perform **webpage entity tagging**, where crucial entities are extracted and categorized from webpage content to ensure that key information is captured. Next, we focus on **query reflection**, anticipating potential user queries by analyzing entity relationships and the webpage's context. This step ensures that the generated SERP summary addresses the information most relevant to the user's search intent. Finally, using the identified entities and anticipated queries, we proceed with **entity-aware summary generation**. This guides large language models (LLMs) to create summaries that are concise, engaging, and aligned with both the user's intent and the webpage's actual content.

In this work, we fine-tune the LLaMa3.1-8B model on entity-aware summaries generated through this framework, optimizing for SERP display. To further improve query-document relevance, we apply Direct Preference Optimization (DPO [16]), ensuring the generated summaries not only reflect webpage entities but also align with likely user queries. Additionally, we enhance system performance through data augmentation techniques, enriching input data with additional context from both search results and LLM-generated contents. The details of these techniques are discussed in Section 3.2.

We introduce an entity-aware summarization framework that improves the relevance and reliability of AI-generated summaries for sponsored search results. By fine-tuning LLaMa3.1-8B using the entity-rich summaries and applying DPO to optimize query-document relevance, the generated summaries exhibit overall F1 score improvements of 82.31% on entity coverage, resulting in a 7.86% increase in recall@50 and an average improvement of 9.75% across all metrics in retrieval tasks. These experimental results highlight significant improvements in entity coverage, retrieval performance, and overall reliability of AI-generated SERP summaries, underscoring the practical impact of our entity-aware approach.

## 2 Method

### 2.1 Preliminaries

**Direct Preference Optimization.** DPO [16] is one of the most popular offline preference optimization methods. In comparison with RLHF [15], DPO eliminates the need for training an additional reward model, instead deriving an implicit reward from the KL-constrained reinforcement learning objective:

$$\max_{\pi_\theta} \mathbb{E}_{x \sim D, y \sim \pi_\theta(y|x)}[r(x,y)] - \beta \mathbb{D}_{\text{KL}}\left[\pi_\theta(y \mid x) \| \pi_{\text{ref}}(y \mid x)\right] \tag{1}$$

$$r_{\text{DPO}}(x,y) = \beta \log \frac{\pi_\theta(y \mid x)}{\pi_{\text{ref}}(y \mid x)} + \beta \log Z(x), \tag{2}$$

where $\pi_\theta$ denotes the policy model and $\pi_{\text{ref}}$ represents the reference model, typically set as the supervised fine-tuned (SFT) model. By incorporating $r_{\text{DPO}}(x,y)$ into the Bradley-Terry (BT) ranking model [2], given an input $x$ from the corpus $D$ with its corresponding preferred output $y_w$ and less preferred output $y_l$, DPO is optimized by minimizing the following loss function:

$$\mathcal{L}_{\text{DPO}}\left(\pi_\theta; \pi_{\text{ref}}\right) = -\mathbb{E}_{(x,y_w,y_l)\sim D}\left[\log \sigma\left(\beta \log \frac{\pi_\theta\left(y_w \mid x\right)}{\pi_{\text{ref}}\left(y_w \mid x\right)} - \beta \log \frac{\pi_\theta\left(y_l \mid x\right)}{\pi_{\text{ref}}\left(y_l \mid x\right)}\right)\right]. \tag{3}$$

In this work, we employ DPO to fine-tune the summarization model to align with search queries, enabling its generations to better bridge the gap between webpages and corresponding queries.

**Dense Passage Retrieval.** Given a query $q$, this task seeks to identify a ranked list of $k$ most relevant passages $\mathcal{L} = [d_1, d_2, \cdots d_k]$ based on the relevance scores from the retrieval model, where $d$ denotes a text passage from the retrieval corpus. In dense retrieval, the queries and passages are both represented in dense vectors, where the relevance score is often computed according to some similarity measurements between the query and passage vectors [6, 10, 13, 21]:

$$\text{Rel}(q, d) = \text{Sim}\big(\text{Enc}_{\text{q}}(q), \text{Enc}_{\text{d}}(d)\big), \tag{4}$$

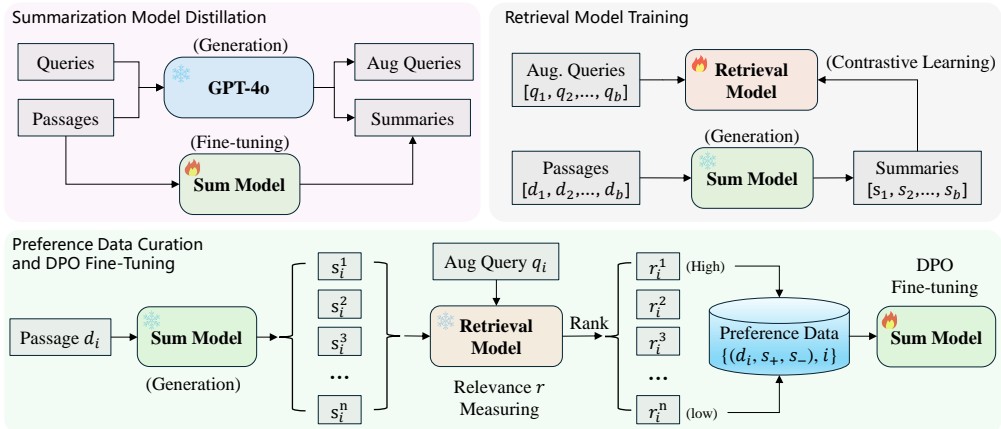

Figure 1: Entity-aware summarization and query-passage relevance in the joint training process: (1) Distilling a summarization model by using GPT-4o to generate entity-aware summaries; (2) Training a retrieval model with augmented queries and the generated summaries; (3) Collecting preference data using the retrieval model and further fine-tuning the summarization model with DPO.

where Enc is the encoding function to map text inputs into dense vectors. In our work, we employ the BERT-base [3] model as the encoder for encoding both queries and passages.

## 2.2 Overview

We provide details on using LLMs to generate entity-aware summaries in Section 2.3, and the training pipeline for the summarization model in Section 2.4.

## 2.3 Entity-aware Summarization Generation

Our framework addresses the challenge of creating concise, informative summaries for sponsored search results by focusing on key entities and potential user intents. It consists of three main steps:

◇ **Webpage Entity Tagging:** Extracts crucial entity information from the webpage content.

    a) **Input:** Raw webpage content W (e.g., DisplayURL, DocumentTitle, Heading, Paragraph, PrimaryContent)

    b) **Process:** LLM identifies and extracts entity types (E_brand, E_product, E_features, E_price, E_audience)

    c) **Output:** Structured list of identified entities E, each tagged with its corresponding type

This step ensures key information is accurately captured for subsequent use. For example, given a webpage about Nike Air Max 90 shoes, it might extract: E_brand: Nike, E_product: Air Max 90, E_features: cushioned sole, breathable mesh, E_price: $120, E_audience: runners, casual wearers.

◇ **Query Reflection:** Generates diverse potential search queries and assesses entity coverage.

    a) **Input:** Set of identified entities E, Original webpage content W

    b) **Process:** Generate 10 diverse possible search queries first, and assess whether extracted entities E contain all information necessary to address these 10 queries

    c) **Output:** List of 10 potential search queries Q

This step ensures the final summary's relevance to various user intents. Continuing the Nike example, it might generate queries like "comfortable running shoes," "Nike latest sneaker release," and "retro style athletic footwear," then assess if the extracted entities sufficiently cover these diverse queries.

◇ **Entity-aware Summary Generation:** Produces a concise, entity-rich summary capturing key information.

a) **Input:** Tagged entities E, Query reflection outputs Q, Original webpage content W

b) **Process:** Generate summary following guidelines (begin with E_brand, mention E_product, include key E_features, etc.)

c) **Output:** Formatted, entity-aware summary of approximately 20-30 words, with entity-aware highlighting

This step synthesizes information to create a tailored summary. For the Nike example, it might produce: "Nike's Air Max 90 offers superior comfort and iconic style for runners and casual wearers. Priced at $120, it features a **cushioned sole and breathable mesh design**." This summary incorporates key entities and addresses diverse potential queries. Detailed prompts for each step can be found in Appendix D.

## 2.4 Training Pipeline for Summarization Model

As illustrated in Figure 1, our pipeline can be roughly divided into three parts: (i) Fine-tuning for Entity-Aware Summarization: We fine-tune LLaMa3.1-8B on GPT-4o generated entity-aware summaries of webpages. (ii) Dense Retrieval Model Training: Using the fine-tuned summarization model, we generate entity-aware summaries for all webpages in our dataset. These summaries, along with their corresponding queries, are then used to train a dense retrieval model. (iii) Summary Refinement via Preference Learning: We employ the summarization model to generate multiple summaries for each webpage. The trained retrieval model is then used to compute similarity scores between these summaries and their corresponding queries. Based on these scores, we identify preferred and less preferred summaries for each webpage. This preference data is utilized to further fine-tune the summarization model using Direct Preference Optimization (DPO). This post-training process enhances the alignment between entity-aware summarization and search queries in downstream retrieval tasks.

To optimize the effectiveness of DPO fine-tuning, we further develop a more diverse preference training set, where the detailed data curation and experimental results are discussed in Section 3.2 and 3.3.3, respectively.

# 3 Experiments

## 3.1 Base Models and Implementation Details

We employ LLaMA-3.1 8B [4] for fine-tuning on generating entity-aware summaries. For dense retrieval, we adopt SimCSE [8] as the training paradigm and employ the BERT-base [3] model as the backbone encoder for encoding both augmented queries and entity-aware summaries.

During initial summarization model fine-tuning, we set the global batch size to 128 and learning rate to 5e-6. For retrieval model training, we use a global batch size of 1024 and a learning rate of 1e-4, with queries limited to 64 tokens and summaries (or raw webpage text) to 448 tokens. The retrieval model is trained for 12k iterations, utilizing a warm-up ratio of 0.1. In DPO training, we set the batch size to 256 and the learning rate to 8e-6, utilizing DeepSpeed [17] Zero Stage 3 for distributed training. Throughout all three training procedures, we adopt a cosine learning rate scheduler and set the first 3% steps as warm-ups, utilizing the AdamW [14] optimizer. All the experiments in this study are conducted on 8 H100-80GB GPUs.

## 3.2 Dataset Curation

**Webpages and Queries**. We crawled in total 100 million webpages from a major commercial search engine, each of which is associated with one distinct user search query. Next, we extract a subset of the collected webpages to create the retrieval training and evaluation benchmark. Specifically, we exclude webpages that do not meet a defined content length range, removing those that are either excessively long or too short. Subsequently, we utilize a fastText-based [11] quality evaluation model[3] to assess the text quality of the remaining webpages, finally selecting 5 million high-quality samples for training, 100k for validation, and 100k for testing.

---

[3]https://huggingface.co/kenhktsui/llm-data-textbook-quality-fasttext-classifier-v2

| Text | Recall | | | nDCG | | MRR | | Avg. |
|------|--------|--------|--------|--------|--------|--------|--------|------|
| | @50 | @200 | @500 | @50 | @200 | @10 | @50 | |
| Webpage | 76.84 | 89.80 | 94.43 | 41.61 | 43.78 | 31.87 | 32.94 | 58.75 |
| Base-Sum | 83.78 | 93.41 | 96.57 | 54.37 | 55.85 | 45.49 | 46.38 | 67.98 |
| EA-Sum | 84.45 | 93.80 | **96.72** | 54.60 | 56.03 | 45.58 | 46.49 | 68.24 |
| Aug-Sum | **84.64** | **93.89** | 96.69 | **54.84** | **56.27** | **45.83** | **46.73** | **68.41** |

Table 1: Retrieval performance across different textual inputs of the webpage. The Webpage, Base-Sum, EA-Sum and Aug-Sum refer to the original webpage text, basic summary, entity-aware summary, and augmented entity-aware summary, respectively.

**Entity-aware Summaries**. We collected 20k GPT-4o [1] generated summaries for each of the basic, entity-aware, and final augmented settings to fine-tune the corresponding summarization models. Details for employing GPT-4o to generate entity-aware summaries are presented in Section 2.3.

**Preference Dataset**. For the preference pairs in DPO fine-tuning, we first select 20k samples from the 5 millon raw webpages with the highest text quality. From each webpage, 16 summaries are sampled using a temperature of 0.9, and the one with the highest similarity to the query is chosen as preferred, while the lowest one is designated as less preferred. These 20k preference pairs create the base preference dataset, referred to as DPO-Base.

To fully leverage the potential of DPO for aligning the generated summaries with queries, we develop a more sophisticated preference dataset with a broader range of preferences. Specifically, we define three types of preference: (i) GPT-4o generated demonstration should be preferred over all generations from the fine-tuned summarization model; (ii) the highest-scoring summary generated from the summarization model should be preferred over any random selection from the remaining ones; and (iii) the preference setting from DPO-Base (highest v.s. lowest). We aggregate 40k, 20k, and 20k preference pairs from these three types, respectively, yielding a total of 80k samples, referred to as DPO-Div, to enhance the fine-tuning of the initial model for alignment with labeled queries.

## 3.3 Main Results

In this section, we discuss entity coverage across different types of summarization and further analyze retrieval performance using various textual representations of the webpage.

### 3.3.1 Entity Coverage in Summarization

We first obtain the labeled entities for each webpage through an existing workflow. Then, we apply an off-the-shelf named entity recognition (NER) model to the generated summaries to extract entities, and we calculate F1 scores for three categories: Brand, Product, and Feature. The results of this entity coverage evaluation are presented in Table 2. The base summary model, which is trained on general summary data, demonstrates limited capability in preserving key entities within the summary. In contrast, our entity-aware summary significantly improves entity retention, capturing 2.2 times more Brands and 2.1 times more Products compared to the base model. Additionally, augmenting the entity-aware summary with webpage category information and a potential query list further enhances entity coverage, particularly for Products and Features, by 19% and 30%, respectively, achieving the best performance overall.

| Summary | Brand | Product | Feature |
|---------|-------|---------|---------|
| Base-Sum | 0.26 | 0.21 | 0.22 |
| EA-Sum | **0.58** | 0.37 | 0.20 |
| Aug-Sum | 0.57 | **0.44** | **0.26** |

Table 2: F1 scores of entity coverage for three types of summaries.

| Model | Recall | | | nDCG | | MRR | | Avg. |
|---|---|---|---|---|---|---|---|---|
| | @50 | @200 | @500 | @50 | @200 | @10 | @50 | |
| SFT | 84.64 | 93.89 | 96.69 | **54.84** | 56.27 | 45.83 | 46.73 | 68.41 |
| DPO-Base | 84.54 | 93.94 | 96.70 | 54.61 | 56.06 | 45.57 | 46.47 | 68.27 |
| DPO-Div | **84.70** | **93.96** | **96.80** | 54.64 | **56.31** | **46.19** | **46.90** | **68.50** |

Table 3: Performance comparison of augmented summaries generated by the summarization model before and after DPO fine-tuning for webpage retrieval.

### 3.3.2 Dense Passage Retrieval

We conducted an extensive evaluation of retrieval performance across four different textual inputs: the original webpage (Webpage), a basic summary (Base-Sum), an entity-aware summary (EA-Sum), and an augmented entity-aware summary (Aug-Sum). The results are summarized in Table 1, measuring standard retrieval metrics, including Recall, nDCG, and MRR at various ranks.

The analysis of different data representations revealed significant variations in retrieval performance. For raw webpages, particularly those featuring products or services, multiple entities are typically scattered across various locations, like the example in Appendix C. Simply extracting raw text from these pages results in the loss of spatial relationships between products. This unstructured format appears noisy and challenging for a small encoder model to process effectively. In contrast, the base summary representation demonstrated a marked improvement in retrieval metrics, increasing from 58.75 to 67.98 for average score. This enhancement can be attributed to the more dense and structured nature of the summarized content, which likely facilitates better learning by the model.

Further gains were observed with the introduction of entity awareness and relevant query augmentation, resulting in an additional 6.3% averae score improvement. This advancement stems from our framework's ability to capture all key entities potentially related to diverse user intents, ensuring a more comprehensive and accurate retrieval process.

### 3.3.3 DPO Preference Fine-tuning

To better align the summarization model with the goals of the downstream webpage retrieval task, we further fine-tuned it using Direct Preference Optimization (DPO) with the preference pairs described in Section 3.1. Table 3 presents the experimental results.

Our findings reveal that constructing preference pairs based solely on query similarity (selecting the summary with the highest similarity as preferred and the lowest as less preferred) does not improve the summarization and retrieval model. However, we found a more effective approach by augmenting the dataset. Specifically, we enhanced the diversity of the preference pairs, creating the DPO-Div dataset. Fine-tuning the summarization model with DPO using this augmented dataset proved more beneficial for the downstream retrieval task. The improvements were particularly notable in the MRR@$k$ metric: MRR@10 improves by 0.62% and MRR@50 improves by 0.43%.

These results suggest that incorporating diverse preference pairs in the DPO fine-tuning process can lead to meaningful improvements in the model's performance on webpage retrieval tasks.

## 4 Conclusion

We propose an entity-aware summarization framework for improving the reliability of AI-generated summaries in sponsored search. Our approach involves a three-step process: webpage entity tagging, query reflection, and entity-aware summary generation. We fine-tune LLaMA3.1-8B on entity-rich summaries and apply Direct Preference Optimization to align generated content with both webpage information and user queries. Extensive experiments demonstrate that our framework significantly improves entity coverage and retrieval performance, outperforming existing summarization approaches for sponsored search. Our method achieves 82.31% higher average F1 scores for Brand, Product, and Feature entities respectively, and shows a 9.75% improvement across all metrics in retrieval tasks.

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

| Text | Recall | | | nDCG | | MRR | | Avg. |
|---|---|---|---|---|---|---|---|---|
| | @50 | @200 | @500 | @50 | @200 | @10 | @50 | |
| Raw | 83.24 | 92.85 | 95.89 | 53.80 | 55.27 | 44.91 | 45.80 | 67.39 |
| Aug. | **84.64** | **93.89** | **96.69** | **54.84** | **56.27** | **45.83** | **46.73** | **68.41** |

Table 4: Performance comparison on query representations. The webpage representations are setting as the augmented summaries.

## A  LLM-based Summarization for Sponsored Search

Recent work in LLM-based summarization for sponsored search has shown promise in generating engaging ad content [9]. Dubey et al. [5] demonstrated improved click-through rates and personalization, respectively, but didn't address entity alignment between summaries and webpage content. Wang et al. [20] advanced keyword generation, while Chang et al. [9] showed LLMs' potential for dynamic content summarization. However, these approaches didn't tackle the crucial issue of misalignment between generated content and source information, a problem highlighted by Song et al. [19] in their work on enhancing content reliability. Our entity-aware summarization framework builds upon these foundations, addressing their limitations by explicitly focusing on key entities, incorporating query reflection, and employing fine-tuning techniques to improve both relevance and accuracy in the sponsored search domain.

## B  Training Data for Basic Summarization Model

This section details the curation of the training data used to fine-tune the basic summarization model, as reflected in the first row of Table 1. Rather than prompting GPT-4o to generate basic demonstration summaries for fine-tuning, we retrieve a dataset of search webpage-related content from the open-sourced CNN/Daily [18]. Specifically, we train a fastText-based [11] classifier using 50k random webpages as positive samples and 50k Wikipedia [7] entries as negative samples. Following previous work [12], we set the model dimension to 256, learning rate to 0.1, the maximum word n-gram length to 3, the minimum word occurrence to 3, and the epoch to 3. Next, we use the trained classifier to select the highest-scoring samples from CNN/Daily, applying a length constraint to exclude samples that significantly deviate from the standard length of our collected webpages. Finally, we obtained 20k webpage-similar samples to train the basic summarization model, mirroring the number of GPT-4o demonstrations used in entity-aware summarization training for a fair comparison.

## C  Example of Entity-aware Summaries and Augmentations

We present an example of a webpage with its corresponding search query, augmented query, basic summary, entity-aware summary, and final augmented summary, as illustrated in Figure 2. To augment the search query, we prompt GPT-4o to infer the user's intent behind the query and attach the title of the first webpage from the search engine following the intent generated by GPT-4o. The experimental results for query augmentation in retrieval tasks are present in Table 4.

In the case of summarization, we find that fine-tuning LLaMA-3.1 8B on open-source summarization datasets fails to effectively capture key entities of the noisy webpage contents, resulting in a simplistic conclusion of each sentence within the webpage. In contrast, fine-tuning the summarization model on entity-aware summaries generated by GPT-4o results in improved summary generation that effectively extracts key entities and identifies their relationships. Furthermore, once trained on augmented entity-aware summaries, the summarization model can follow this pattern to generate similar augmented summaries, which encompass both webpage classification and potential queries related to the webpage.

## D  GPT-4o Prompts for Entity-aware Summary Generation

The step-wise prompts for GPT-4o to generate the entity-aware summaries are presented in Table 3.

**Raw text**:
www boatoutfitters com crownline boat parts [SEP] crownline boat parts replacement parts for crownline boats [SEP] stormtrooper deluxe child and teen costume mandalorian deluxe child costume yoda child costume darth vader deluxe child costume mandalorian the child baby and toddler costume deluxe darth vader adult costume chewbacca adult costume deluxe stormtrooper adult costume stormtrooper lady adult costume darth vader deluxe adult costume princess leia adult costume deluxe sith trooper child costume darth vader child and teen costume boba fett child costume chewbacca child costume darth maul child and teen costume princess leia child and teen costume jedi robe child costume star wars clone trooper blaster 45cm boba fett blaster 62cm stormtrooper blaster 38cm lightsaber 66cm [SEP] hardware material other parts can t find it we build thousands of boat parts not listed online call or send a picture we d love to help track it down we ve supplied crownline boats with oem component parts used on new production boats since 1993 in those nearly 30 years of working together we ve prototyped and brought into production over 1 150 unique crownline parts including doors tables storage boxes substrate panels and much more this relationship allows us to provide crownline boat owners with the largest selection of crownline replacement hardware material exact replacement oem parts crownline primarily utilizes white white king starboard and 7508 acrylic accessories as well as black acrylic accents they ve used a number of different solid surface colors over the years depending on the model year and replacement part you need a sample chain may be necessary to guarantee color match [SEP] home parts crownline replacement parts sort by relevance select options add to cart showing 22 of 22 your source for hard to find crownline boat parts shop exact replacement crownline parts directly from a long term crownline supplier.

---

**Raw Query**:
rownline bow filler cushions

**Augmented Query**:
rownline bow filler cushions

*User Intent*: The user is looking for bow filler cushions that are compatible with Crownline boats. Bow filler cushions are soft pads that can be inserted between the front seats of a boat to create a larger and more comfortable lounging area

*First title of web search result*: Original Crownline Boat Parts Online Catalog - Great Lakes Skipper

---

**Basic Summary**:
Crownline boat parts replacement parts for crownline boats. Stormtrooper deluxe child and teen costume. Mandalorian deluxe child costume. Yoda child costume. Darth vader deluxe child costume. Mandalorian the child baby and toddler costume. Deluxe darth vader adult costume. Chewbacca adult costume. Deluxe stormtrooper adult costume. Stormtrooper lady adult costume. Darth vader deluxe adult costume. Princess leia adult costume. Deluxe sith trooper child costume . Darth vader child and teen costume . Boba fett child costume. Darth maul child and teen costume. Princess leia child and teen costume. Star wars clone trooper blaster 45cm. Boba fett blaster 62cm. Stormtrooper blaster 38cm.

**Entity-aware Summary**:
**Boat Outfitters** offers a wide range of **Crownline boat parts** including **OEM components** and **replacement hardware** with a focus on **exact matches** and **color consistency** for **Crownline boat owners**.

**Augmented Entity-aware Summary**:
*Webpage category*: crownline parts | Boat Parts and Accessories

*Entity-aware summary*: **Crownline** offers **replacement parts** for **Crownline boats**, including **OEM components** and **custom-built parts** to ensure exact color matches and a wide selection of options.

*10 potential user queries*: crownline boat parts, crownline replacement parts, crownline OEM parts, crownline boat hardware, crownline boat accessories, crownline boat material, crownline boat storage boxes, crownline boat tables, crownline boat doors, crownline boat substrate panels

Figure 2: An example of a webpage text with its corresponding query, augmented query, basic summary, entity-aware summary, and augmented entity-aware summary. During generating augmented contents and training the retrieval model, all augmented texts are concatenated using [SEP].

**1. Webpage Entity Tagging**:
<|im_start|>system
Identify crucial elements from the webpage content:
- Product name or primary service
- Brand or retailer
- Key features, ingredients, or benefits (2-3 most important)
- Price or price range (if available)
- Target audience, use case, or occasion (if specified)
- Unique selling points or distinguishing factors

Apply entity-aware formatting:
Brand: Brand
Product: Product
Product Models: Product_Models
Services: Services
Features: Features
Audience: Audience
Price: Price
<|im_end|>

---

**2. Query Reflection**:
<|im_start|>system
Generate 10 search queries:
Provide 10 search queries that could trigger relevant ads for the given webpage within <Query> </Query> tags.

Determine ad category:
Specify the ad category of the given webpage within <Category></Category> tags.
<|im_end|>

---

**3. Entity-Aware Summarization**:
<|im_start|>system
Create a concise, single-sentence summary:
- Begin with the brand name or retailer
- Mention the product name or primary service
- Include 2-3 key features, ingredients, or benefits
- Add price information if available and relevant
- Mention target audience, use case, or occasion if applicable
- Highlight any unique selling points or distinguishing factors
- Aim for approximately 20-30 words

Apply entity-aware formatting in the summary:
Use text formatting ** to highlight:
- Brand or retailer in blue
- Product name or primary service in red
- Key features, ingredients, or benefits in green
- Price in orange
- Target audience, use case, or occasion in purple
- Unique selling points in bold

Provide reasoning:
Explain your choices for the summary and include identified entities within <Reasoning></Reasoning> tags.

Present the final summary within <Summary></Summary> tags.
<|im_end|>

Figure 3: Prompts for GPT-4o to generate the entity-aware summaries, including instructions for extracting crucial entities from webpages, generating additional queries relevant to the webpage, classifying the webpage, and ultimately providing the entity-aware summary with explanations.

