# OpenReview forum: "What You See Is What You Get: Entity-Aware Summarization for Reliable Sponsored Search"
_NeurIPS.cc/2024/Workshop/SafeGenAi — SafeGenAi Poster_

### Official Review · Reviewer_UKFY · 2024-10-08
**Paper Review: "What You See Is What You Get: Entity-Aware Summarization for Reliable Sponsored Search"**

**Rating:** 7
**Confidence:** 3

**Review:**

Overview
The paper introduces an entity-aware summarization framework aimed at enhancing the reliability of AI-generated summaries in sponsored search contexts. The authors address the common issue of misalignment between AI-generated summaries and actual webpage content, which can mislead users and affect the accuracy of information retrieval. The proposed method involves a structured process for generating entity-aware summaries and fine-tuning a language model (LLaMa3.1-8B) with Direct Preference Optimization (DPO) to improve query-document relevance.
Pros
Innovative Approach:
The paper proposes a novel entity-aware summarization framework that focuses on improving the alignment between generated summaries and actual webpage content, which is crucial for sponsored search environments.
Comprehensive Methodology:
The approach includes detailed steps like webpage entity tagging, query reflection, and summary generation, ensuring that the summaries are both relevant and informative.
Fine-tuning the LLaMa3.1-8B model on entity-rich summaries and applying DPO enhances the relevance of these summaries to user queries, which is a significant advancement over traditional summarization methods.
Strong Empirical Results:
The paper reports substantial improvements in F1 scores for entity recognition in summaries and shows a notable increase in recall@50 for retrieval tasks, indicating that the model performs well on both summarization and retrieval fronts.
Detailed Evaluation:
Extensive experiments and comparisons with baseline models provide a clear picture of the framework's effectiveness. The use of metrics like F1 scores and recall@50 helps in quantitatively assessing the performance improvements.
Cons
Complexity in Implementation:
The multi-step process involving entity tagging, query reflection, and DPO might be complex to implement and optimize in practical, real-world systems.
Potential for Bias:
There is always a concern with AI-generated content regarding the potential introduction of biases, especially when dealing with entity recognition and summarization. The paper does not address how it mitigates potential biases in generated summaries.
Conclusion
The paper presents a significant contribution to the field of AI-driven summarization for sponsored search, offering a robust framework that improves the accuracy and relevance of summaries. Despite the challenges related to implementation complexity and computational demands, the proposed method's ability to enhance user trust and satisfaction in sponsored search results is commendable. Further research could explore the framework's applicability in other domains and address potential bias in AI-generated content.

---

### Official Review · Reviewer_TwAf · 2024-10-09

**Rating:** 6
**Confidence:** 4

**Review:**

**Summary**

This paper introduces an entity-aware summarization framework aimed at improving the reliability of AI-generated summaries for sponsored search results. The framework consists of a structured, three-step process involving webpage entity tagging, query reflection, and entity-aware summary generation. It fine-tunes the LLaMa3.1-8B model on entity-rich summaries and applies Direct Preference Optimization (DPO) to optimize query-document relevance. Comprehensive evaluations demonstrate the effectiveness of the proposed method in improving entity coverage and retrieval performance across multiple benchmarks. The framework achieves significant improvements in F1 scores for various entities and boosts recall in retrieval tasks.

**Strength**

- The paper addresses a crucial issue in sponsored search—misalignment between AI-generated summaries and actual webpage content. It presents a structured, step-by-step solution that is both well-motivated and practically relevant.
- The experimental setup is robust, with extensive evaluations across multiple retrieval and summarization tasks. The improvement in F1 scores for entity coverage and retrieval metrics such as recall and nDCG demonstrate the efficacy of the proposed approach.
- The paper provides detailed explanations of key concepts, including entity-aware summarization, query reflection, and the role of DPO in optimizing query-document relevance. The inclusion of prompts used for GPT-4o is helpful for understanding the process.
- Sponsored search is a critical component of many commercial search engines, and improving the accuracy and relevance of AI-generated summaries in this context has significant real-world implications. The proposed framework addresses a key pain point for both advertisers and users, making it a valuable contribution.

**Weakness**

- The pipeline involves multiple stages, including entity tagging, query reflection, and DPO fine-tuning. While each step is well-documented, the complexity of the entire pipeline may pose challenges for practical deployment at scale.
- The paper primarily focuses on sponsored search, and while this is an important application, the broader applicability of the framework to other types of information retrieval tasks remains unexplored.

---

### Official Review · Reviewer_8PJv · 2024-10-09
**Innovative summarization framework significantly enhances AI-generated summary alignment with web content, demonstrating notable improvements in recall and F1 scores.**

**Rating:** 8
**Confidence:** 4

**Review:**

The paper introduces an innovative framework for generating reliable AI-driven summaries for sponsored search results. By integrating webpage entity tagging, query reflection, and summary generation, and enhancing these with a fine-tuned LLaMa3.1-8B model using Direct Preference Optimization (DPO), the authors significantly improve the alignment of summaries with actual webpage content. The method demonstrates substantial improvements over baselines in both recall and F1 scores for entity recognition. However, the paper could benefit from exploring the scalability of DPO with complex queries and testing on more diverse datasets to confirm the method's robustness and applicability in varied real-world scenarios.